# Effects of paleogeographic changes and $CO_2$ variability on northern mid-latitudinal temperature gradients in the Cretaceous

Kaushal Gianchandani [1] ✉, Sagi Maor [1], Ori Adam [1], Alexander Farnsworth [2,3], Hezi Gildor [1], Daniel J. Lunt [2] & Nathan Paldor [1]

The Cretaceous 'greenhouse' period (~145 to ~66 million years ago, Ma) in Earth's history is relatively well documented by multiple paleoproxy records, which indicate that the meridional sea surface temperature (SST) gradient increased (non-monotonically) from the Valanginian (~135 Ma) to the Maastrichtian (~68 Ma). Changes in atmospheric $CO_2$ concentration, solar constant, and paleogeography are the primary drivers of variations in the spatio-temporal distribution of SST. However, the particular contribution of each of these drivers (and their underlying mechanisms) to changes in the SST distribution remains poorly understood. Here we use data from a suite of paleoclimate simulations to compare the relative effects of atmospheric $CO_2$ variability and paleogeographic changes on mid-latitudinal SST gradient through the Cretaceous. Further, we use a fundamental model of wind-driven ocean gyres to quantify how changes in the Northern Hemisphere paleogeography weaken the circulation in subtropical ocean gyres, leading to an increase in extratropical SSTs.

Analysis of oxygen isotopes in foraminifera tests ($\delta^{18}O$) and the index of tetraethers that quantifies the archaeal lipid distributions in marine sediments ($TEX_{86}$) indicate that the sea surface temperature (SST) during the Cretaceous period (~145 to ~66 million years ago, Ma) ranged between 28–35 °C in low latitudes to 11–22 °C in high latitudes (first and third quartiles of LOESS smoothened data)[1]. A similar trend is observed during the Paleocene (~66 to ~56 Ma) and Eocene (~56 to ~34 Ma) epochs of the Paleogene period (~66 to ~23 Ma) when the tropical and high latitude temperatures were between 31–33 °C and 11–14 °C, respectively[2]. The persistence of such high globally averaged SST and low pole-to-tropics temperature gradient during the Cretaceous–Paleocene–Eocene (CPE) is attributed predominantly to the strong greenhouse effect ensuing from the higher partial pressure of $CO_2$ ($pCO_2$) in the atmosphere[3,4]. However, in addition to changes in $pCO_2$, the spatiotemporal variability in SST can also stem from

tectonically driven changes in paleogeography, including surface topography, ocean bathymetry, and the arrangement of continents[5–7]. Specifically, paleogeographic alterations during the Cretaceous have been demonstrated to dampen seasonal variations and trigger widespread flooding, ultimately leading to moderation of the pole-to-tropics temperature gradient[8,9]. The rearrangement of continents during the period has also been associated with shifts in the deep-water formation sites[7,10], alternations in the oceanic overturning circulation[11], and variations in mid-Cretaceous deep-sea ventilation[12]. Furthermore, the paleogeographic features associated with certain ages in the Early and Late Cretaceous have been found to be more conducive to glacial events than the mid-Cretaceous[13]. Moreover, changes in Antarctic orography[14] and the opening of the Drake passage[15] could have both contributed to the variability in meridional SST gradients during the Paleocene–Eocene (PE).

[1]Fredy & Nadine Herrmann Institute of Earth Sciences, Hebrew University of Jerusalem, Edmond J. Safra Campus, Givat Ram, Jerusalem 9190401, Israel. [2]School of Geographical Sciences and Cabot Institute, University of Bristol, Bristol BS8 1SS, UK. [3]State Key Laboratory of Tibetan Plateau Earth System, Environment and Resources (TPESER), Institute of Tibetan Plateau Research, Chinese Academy of Sciences, Beijing 100101, China. ✉e-mail: kaushal.g@mail.huji.ac.il

Although previous studies have investigated the effect of different paleogeographies corresponding to multiple ages in the CPE on variations in the pole-to-tropics temperature gradient and the ocean circulation[10–12,16], the effect of paleogeographic changes on the surface ocean circulation specifically has not been examined before. Surface ocean gyres contained between two meridional continental boundaries are prominent features of the present-day ocean. The meridional volume (mass) transport ($\psi$) associated with these gyres redistributes heat in the ocean and is critical to regulating the climate on Earth's surface.

Henry Stommel was the first to formulate a simple mathematical model comprising five parameters that capture the fundamental characteristics of the circulation in such gyres[17]. While the linear dependence of $\psi$ on the amplitude of zonally averaged wind stress ($\tau$) is evident from the analytical solution of Stommel's model, discerning its dependence on the zonal and meridional extents of the ocean basin ($L_x$ and $L_y$, respectively) is rather non-trivial. It should be noted that in Stommel's model, the meridional boundaries of the basin are co-located with the latitudes of zero wind-stress curl. However, in the real ocean, the two do not necessarily coincide in general, and the extent of the surface gyres is constrained only by the latitudes corresponding to the vanishing wind-stress curl. Previous studies established that both $\psi$ and the poleward heat transport ($\mathcal{H}$) associated with a surface gyre increase with the horizontal aspect ratio $\left(\frac{L_y}{L_x}\right)$ of the ocean basin containing the gyre[18–20] (hereafter referred to as the 'gyral basin'). Thus, the aspect ratio of any gyral basin is governed by the curl of the overlying wind stress and the relative position of the zonal boundaries set by the continents. Based on these first-order constraints, it was recently postulated that the large meridional SST gradients observed during some geologic periods may have resulted from the smaller $\psi$ (and consequently a smaller $\mathcal{H}$) associated with the surface gyres contained in basins with small aspect ratios[20].

Here we use data generated from two ensembles of paleoclimate simulations carried out using the HadCM3L (specifically, HadCM3LBM2.1aD) model[6,7] to better constrain the relative contribution of changes in $p$CO$_2$ and paleogeographic effects on the SST distribution during the CPE. In particular, we develop a simple model that isolates the role of paleogeographically driven changes in extratropical gyral circulation and the corresponding effect on the evolution of pole-to-tropics SST gradient. The model is then applied to the HadCM3L data to show that paleogeography-driven reduction in $\psi$ (and consequently $\mathcal{H}$) associated with the surface ocean gyres may have been a key factor contributing to the variability in meridional SST gradients observed during the CPE[1,2].

## Results
### Numerical simulations
All simulations necessary for this work were carried out using the UK Met Office coupled atmosphere-ocean model, HadCM3L (resolution: 3.75° in longitude × 2.5° in latitude), which includes multiple climate feedbacks, including vegetation feedbacks[6,21]. The atmospheric CO$_2$ concentration in the two ensembles is kept constant at 560 ppmv and 1120 ppmv (i.e., ×2 and ×4 preindustrial atmospheric CO$_2$ concentration) while the geography and the solar constant are varied. The chosen values of atmospheric CO$_2$ provide an idealized framework that allows us to isolate the effect of CO$_2$ versus paleogeography. In addition, this range of CO$_2$ values captures the maximum variation (530 ppmv to 840 ppmv) in the long-term trend of LOESS smoothened data on estimated atmospheric CO$_2$ concentration during the CPE[22]. The range does not capture the very lowest values of CO$_2$ seen in some proxies in the latest Cretaceous[23], but these low values are inconsistent with relatively high reconstructed global mean temperatures at this time[24]. The geographies corresponding to different geological ages are constrained by data aggregated from lithological, tectonic, fossil, and

deep-sea studies[6]. The solar constant in both ensembles increases monotonically from the Berriasian age (~143 Ma) to the Priabonian age (~36 Ma) by 0.9% (see Methods). In practice, the relatively small magnitude of the forcing associated with the increase in solar constant is dwarfed by changes in $p$CO$_2$ and paleogeography between simulations and their relative impact on SST distributions.

### Wind-driven gyral circulation in the CPE
The mid-latitudinal gyral basin in the northern paleo-Pacific (hereafter referred to as the North Pacific) retains an approximately rectangular geometry − similar to Stommel's idealized ocean − during the CPE, despite considerable plate tectonic activity[25,26]. Variations in the North Pacific's circulation are expected to strongly affect the climate during the CPE, since for these ~110 million years much of the planet's seawater was in the large paleo-Pacific[27,28].

The $L_x$ of the mid-latitudinal gyral basin in the North Pacific, i.e., the typical longitudinal distance between continental margins (see Methods), remained nearly constant during the CPE (~13,000 km, Supplementary Fig. 1). On the other hand, $L_y$ of this basin, which is defined as the distance between the two latitudes where the wind-stress curl over the surface vanishes, decreased (non-monotonically) from ~3900 km to ~2800 km and then increased to ~3600 km (Fig. 1a). These variations in $L_y$ determine the variations in the aspect ratio and hence the value of $\psi$ in this gyral basin. This is illustrated by comparing $\psi$ in the longer Valanginian gyral basin (~135 Ma; Fig. 1b), which is double that of the ~30% narrower Maastrichtian gyral basin (~68 Ma; Fig. 1c). The ocean during the Valanginian is also characterized by a much weaker SST gradient, which is consistent with a stronger $\mathcal{H}$ compared to the Maastrichtian.

### Variability in meridional SST gradients and its dependence on atmospheric CO$_2$ and paleogeography
To examine the variability in meridional SST gradients during the CPE, we first consider the SST gradients from the Early to the Late Cretaceous and subsequently focus on the trend from the Paleocene to the Eocene. Specifically, we compare the Maastrichtian to the Valanginian (rather than the Berriasian, given the paucity of the proxy-derived SST data in this age[1]) in the Cretaceous and the Priabonian to the Danian age (~63 Ma) in the PE.

In the HadCM3L simulations carried out for an atmospheric CO$_2$ concentration of 560 ppmv, the meridional gradient of the zonally averaged SST (over all longitudes, $\Delta SST_y$) calculated between 20°N and 50°N increases from 12.5 °C in the Valanginian to 15.7 °C in the Maastrichtian, whereas, $\Delta SST_y$ decreases from 16.5 °C in the Danian to 15.5 °C in the Priabonian. The same trend persists for an atmospheric CO$_2$ concentration of 1120 ppmv, $\Delta SST_y$ increases from 11.1 °C to 13.9 °C during the Cretaceous and decreases from 14.3 °C to 13.7 °C during the PE. Figure 2a illustrates the temporal trend in $\Delta SST_y$ for the two atmospheric CO$_2$ concentrations. In both simulation ensembles, the temporal variation in $\Delta SST_y$ from the Valanginian to the Maastrichtian is ~3 °C even though the global mean SST varies only slightly (standard deviation of less than 1 °C about the ensemble averages, see Supplementary Fig. 3). Alternatively, the temporal variation in $\Delta SST_y$ from the Danian to the Priabonian (~0.8 °C) is comparable to the standard deviation about the global mean SST for the ensembles. Furthermore, comparing the two simulation ensembles shows that during the CPE, a doubling of $p$CO$_2$ decreases $\Delta SST_y$ between 0.7 °C (observed in the Albian age, ~106 Ma) and 2.6 °C (observed in the Selandian age, ~60 Ma), resulting in an average decrease of 1.8 °C in $\Delta SST_y$.

The long-term trend of LOESS smoothened data on estimated atmospheric CO$_2$ concentration suggests that a decrease in atmospheric CO$_2$ from ~700 ppmv in the Valanginian to ~570 ppmv in the Maastrichtian[22] can cause a decrease in global mean SST by ~1.5 °C (based on a typical climate sensitivity parameter of ~4.5 °C per CO$_2$

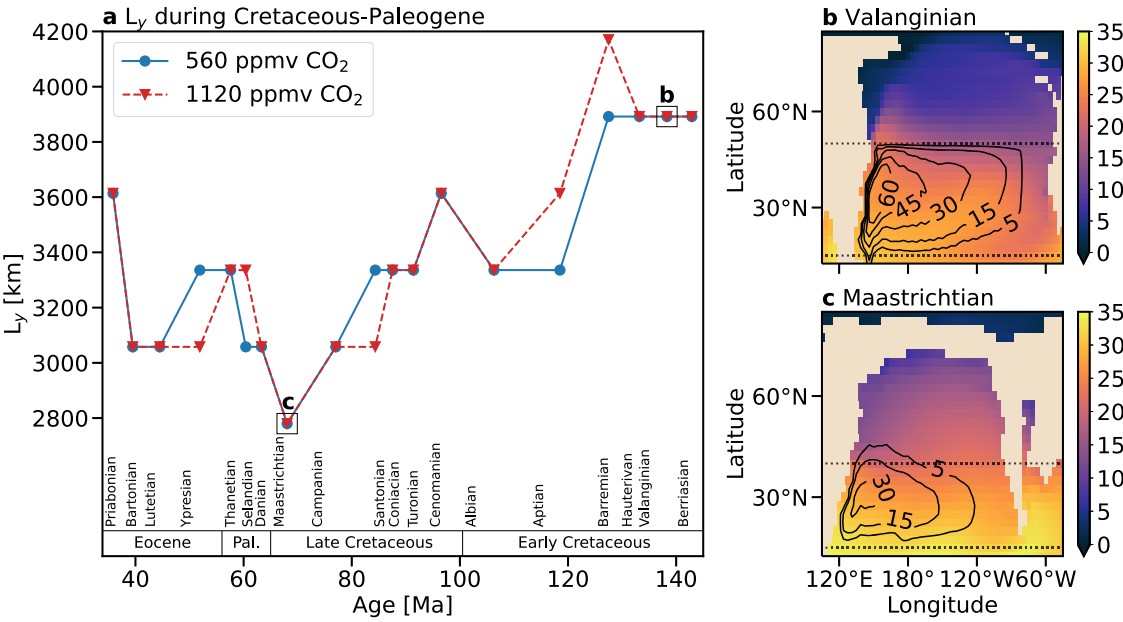

**Fig. 1 | Changes in the meridional extent of the mid-latitudinal gyral basin ($L_y$) in the Northern paleo-Pacific during the Cretaceous and Paleogene periods for an atmospheric $CO_2$ concentration of 560 ppmv (solid blue curve) and 1120 ppmv (dashed red curve). a** Change in $L_y$ during the Early-Late Cretaceous, the Paleocene, and the Eocene epochs. **b, c** Sea surface temperature (°C, color) and gyral streamlines (Sv, contour; $1\,Sv = 10^6\,m^3\,s^{-1}$) in the ocean during the Valanginian (-135 Ma) and the Maastrichtian (-68 Ma) ages for an atmospheric $CO_2$ concentration of 1120 ppmv. Dotted lines mark the latitudes where the wind-stress curl is zero. For completeness, Supplementary Fig. 2 shows the sea surface temperature fields and streamlines in the ocean during the Valanginian and the Maastrichtian for an atmospheric $CO_2$ concentration of 560 ppmv.

doubling[7]) and an increase in $\Delta SST_y$ by -0.6 °C (since a doubling of atmospheric $CO_2$ decreases $\Delta SST_y$ by 1.8 °C a decrease in $CO_2$ from -700 ppmv to -570 ppmv, i.e., a reduction by a factor of 0.8, is expected to increase $\Delta SST_y$ by $\frac{\log(0.8)}{\log(2)} \times -1.8 = 0.6$ °C). Similarly, a decrease in atmospheric $CO_2$ from -590 ppmv in the Danian to -530 ppmv in the Priabonian[22] is expected to cause a 0.7 °C decrease in global mean SST and an 0.3 °C increase in $\Delta SST_y$. These estimates are consistent with the inverse relation between $\Delta SST_y$ and global mean SST observed across multiple climate states in Earth's history with varied paleogeographies[2]. It should be noted that changes in atmospheric $CO_2$ may have had a more pronounced effect on $\Delta SST_y$ since this study does not account for the short-term fluctuations in $pCO_2$ trend or the lowest values of $CO_2$ inferred from some proxies[23].

Comparing the typical variation in $\Delta SST_y$ from the change in $pCO_2$ with paleogeography-driven variation in $\Delta SST_y$ indicates that paleogeographic changes had a greater effect on the mid-latitudinal temperature gradient during the Cretaceous than the long-term changes in $pCO_2$. This is also true for the PE. However, since the overall $\Delta SST_y$ variation during the PE is comparable to the standard deviation in the ensemble-averaged global mean SST, the analysis that follows only focuses on the Cretaceous.

The differences between the long-term SST gradients in the simulations and estimates based on paleotemperature reconstructions may be attributed to the limited spatial distribution of available proxies. Paleotemperature reconstructions based on $\delta^{18}O$ and $TEX_{86}$ proxies suggest that meridional SST gradients first decreased from -10–17 °C in the Valanginian to -3–5 °C in the Aptian (-119 Ma) and subsequently increased through the middle–Late Cretaceous to reach a maximum of -19–21 °C in the Maastrichtian[1]. However, the trend in this proxy-derived meridional SST gradient is only expected to capture the long-term trends in Equator-to-pole temperature difference. It does not capture the mid-latitudinal temperature gradient in the Northern Hemisphere (the focus of this study) since no data exists between 30°N and 80°N and the data for the high latitude paleo-temperature originates from sites located between 48°S and 54°S paleolatitudes. Nevertheless, the estimated $\Delta SST_y$ variation resulting

from the combined effect of reduction in atmospheric $CO_2$ concentration and paleogeographic changes from the Valanginian to the Maastrichtian (-3.6 °C) is within 55% of the variation estimated from proxies (-6.5 °C).

## Role of gyral circulation in determining the meridional SST gradients

The decrease in $\Delta SST_y$ resulting from an increase in $pCO_2$ of the model atmosphere can be attributed to polar amplification[22, 29–32], whereas the increase in $\Delta SST_y$ from the Early to Late Cretaceous in the two ensembles of simulations is related to paleogeography-driven changes. Paleogeography can influence many factors that could in turn affect polar amplification (e.g., cloud distribution and other local feedbacks); here, we focus on its influence on oceanic heat transport ($\mathcal{H}$). Figure 2b shows a strong anti-correlation between $\Delta SST_y$ and the maximal volumetric (mass) transport ($\psi$) in mid-latitudes, which indicates that Cretaceous basins with smaller $L_y$ contain surface gyres with smaller $\psi$. Thus, the weakening of mid-latitudinal $\mathcal{H}$ (Supplementary Fig. 4), caused by the weakened mid-latitude ocean gyres contributes substantially to the increase of $\Delta SST_y$ in the middle−Late Cretaceous. To quantify this effect, we assume that:

1. To first order, $\Delta SST_y$ decreases linearly with $\mathcal{H}$ and with the poleward heat transport associated with the atmospheric meridional overturning circulation (MOC).
2. The heat transport associated with the atmospheric MOC is proportional to $\mathcal{H}$[33].
3. The ocean circulation, to first order, is similar to the circulation in a 'Stommel-like' gyre. As such, due to rapid advection, there is negligible loss of heat by warm water along the western boundary. In contrast, the water mass loses much of its heat while it traverses the northern and eastern edges of the basin at a much lower speed. Thus, SST along the southern and western boundaries is high, whereas SST along the northern and eastern boundaries is low. Further, the East−West SST gradient is proportional to $\Delta SST_y$.

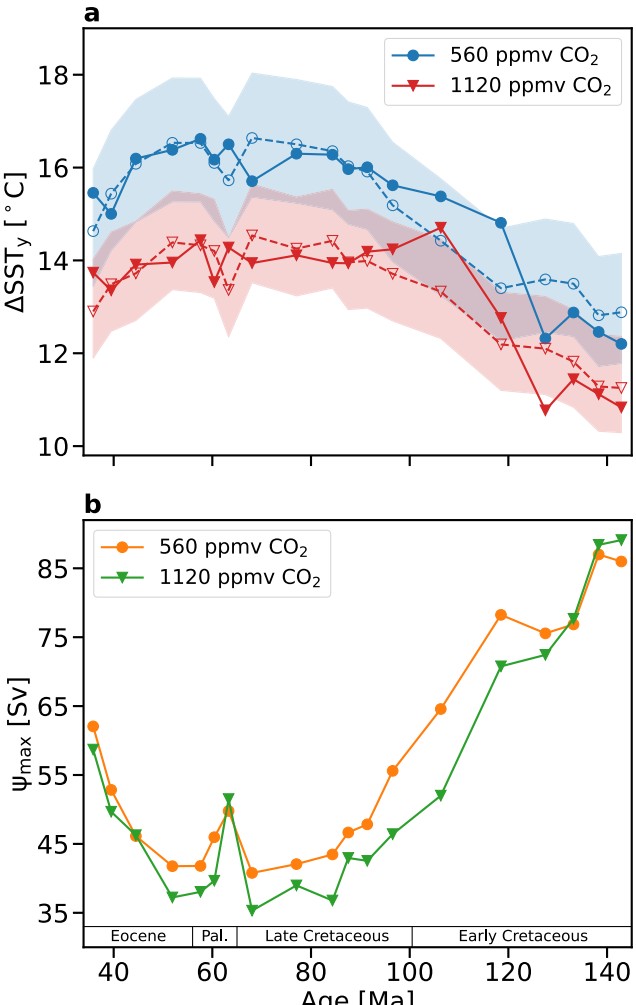

**Fig. 2 | Meridional temperature gradient and maximal volumetric (mass) transport in the mid-latitudinal northern paleo-Pacific. a** Meridional gradient of zonal-mean SST ($\Delta SST_y$) between 20°N and 50°N for an atmospheric $CO_2$ concentration of 560 ppmv (blue curve) and 1120 ppmv (red curve). The best-fit curves for the two atmospheric $CO_2$ concentrations (dashed blue and red curves) are calculated using $\Delta SST_y$ and the maximal volume (mass) transport ($\psi_{max}$). The shaded region shows the range between which the best-fit curve can vary with 1σ error in $\kappa$ and $\Delta SST_y^{Rad}$. **b** Decrease in $\psi_{max}$ from the Early to Late Cretaceous and the subsequent increase from the Paleocene to the Eocene for the two atmospheric $CO_2$ concentrations.

Based on these assumptions, we find (see Methods):

$$\Delta SST_y = \frac{\Delta SST_y^{Rad}}{1 + \left(\kappa \times \psi_{max} \times \Delta SST_y^{Rad}\right)} \quad (1)$$

where $\Delta SST_y^{Rad}$ is the meridional gradient of SST at radiative equilibrium ($\mathcal{H} = 0$), $\psi_{max}$ is the maximal gyral volume (mass) transport in the mid-latitudinal North Pacific, and $\kappa$ is an empirical constant. The above equation is developed based on the paleogeographic considerations relevant to the Cretaceous, but we expect it to be also applicable to the PE since the majority (>60%) of the volumetric (mass) transport in the ocean's surface is concentrated in the North Pacific until the Priabonian. Performing a regression analysis, based on this equation, on $\Delta SST_y$ and $\psi_{max}$ estimated from the HadCM3L model data for the 19 ages considered here yields $\Delta SST_y^{Rad} = 22.6 \pm 1.4$ °C and $\kappa = (3.9 \pm 0.5) \times 10^{-4}$ Sv$^{-1}$ °C$^{-1}$ (1 Sv $= 10^6$ m$^3$ s$^{-1}$) for an

atmospheric $CO_2$ of 560 ppmv, and $\Delta SST_y^{Rad} = 18.0 \pm 0.9$ °C and $\kappa = (3.7 \pm 0.6) \times 10^{-4}$ Sv$^{-1}$ °C$^{-1}$ for an atmospheric $CO_2$ of 1120 ppmv. The best-fit curves (Fig. 2a, dashed curves) illustrate that the reduction in $\psi_{max}$ explains ~80% and ~75% of the variance in $\Delta SST_y$ from the Berriasian to the Priabonian in the simulation ensemble with an atmospheric $CO_2$ of 560 ppmv and 1120 ppmv, respectively. This underscores the role of $L_y$ in the increase in $\Delta SST_y$ during the Cretaceous and in the subsequent decrease in $\Delta SST_y$ from the Paleocene to the Eocene.

Though our regression analysis captures the general trend in the evolution of $\Delta SST_y$ during the CPE for both simulation ensembles, we observe a steep increase in $\Delta SST_y$ between ~130 Ma and ~100 Ma for the simulations with an atmospheric $CO_2$ of 1120 ppmv (Fig. 2a, solid red curve). This steep increase lies outside the range in which the best-fit curve is expected to vary based on one standard deviation variation in $\kappa$ and $\Delta SST_y^{Rad}$. It stems from an abrupt decrease in the SST at 50°N between the Barremian age (~128 Ma) and the Albian age (Supplementary Fig. 5). However, examining the precise climatic variations and feedback associated with paleogeographic changes that led to this abrupt cooling between the Barremian and the Albian are beyond the scope of this study.

The HadCM3L simulations show that geologic ages in the Cretaceous with a large $L_y$ (e.g., the Valanginian) are characterized by a wide Equator-to-pole extent of the ocean basin ($\tilde{L}_y$) (Fig. 3a, b), while geologic ages with a small $L_y$ (e.g., the Maastrichtian) are marked by a narrower $\tilde{L}_y$ (Fig. 3c, d). Moreover, like $L_y$, $\tilde{L}_y$ also decreased from the Early to Late Cretaceous (Supplementary Fig. 6). The correlation between $L_y$ and $\tilde{L}_y$ can be attributed to the increase in land area in the polar North Pacific since the mid-Cretaceous (Supplementary Fig. 7), because the increased land area amplifies the damping effect (friction) over the surface, which reduces the meridional distance between the zonal wind extremes (thus reducing the distance between the latitudes where the wind-stress curl over the surface vanishes). However, multiple other features of the atmosphere-ocean system not considered here can affect $L_y$, e.g., meridional extent and/or intensity of the Hadley cell, the intensity of the Walker circulation, and exchange of water masses between ocean basins[34]. For instance, $L_y$ and $\tilde{L}_y$ do not coevolve during the PE (Supplementary Fig. 6), which could be related to one or more of these features. Exploring a causal relation between changes in these climatic features and $L_y$ on geologic timescales is left for future study.

## Discussion

In summary, we find that a large portion of the increase in $\Delta SST_y$ observed in the Stommel-like mid-latitudinal North Pacific basin during the Cretaceous results from the decrease in poleward ocean heat transport. This decrease in heat transport is a consequence of a weakened intensity of the gyral circulation, i.e., a reduction in the volumetric/mass transport associated with the gyre. This weakening results from tectonically driven changes in Northern Hemisphere paleogeography, which reduces the horizontal aspect ratio of the gyral basin. The anti-correlation between oceanic heat transport and $\Delta SST_y$ persists in the PE as well, and an increase in oceanic heat transport during these epochs is accompanied by a decrease in $\Delta SST_y$. However, this decrease is of the order of the standard deviation of the ensemble-averaged global mean SST. We note that in contrast to the Northern Hemisphere, the two ensembles of simulations do not show a discernible trend in $\Delta SST_y$ variations in the Southern Hemisphere, likely because of the break-up of the Antarctica–Australia landmass during this period – demonstrating the limitations of the theory proposed here.

We also find that fluctuations in atmospheric $CO_2$ concentration can substantially affect the $\Delta SST_y$ during the Cretaceous and estimate the magnitude of $\Delta SST_y$ variation based on the long-term trend of atmospheric $CO_2$. However, there is a considerable uncertainty in the

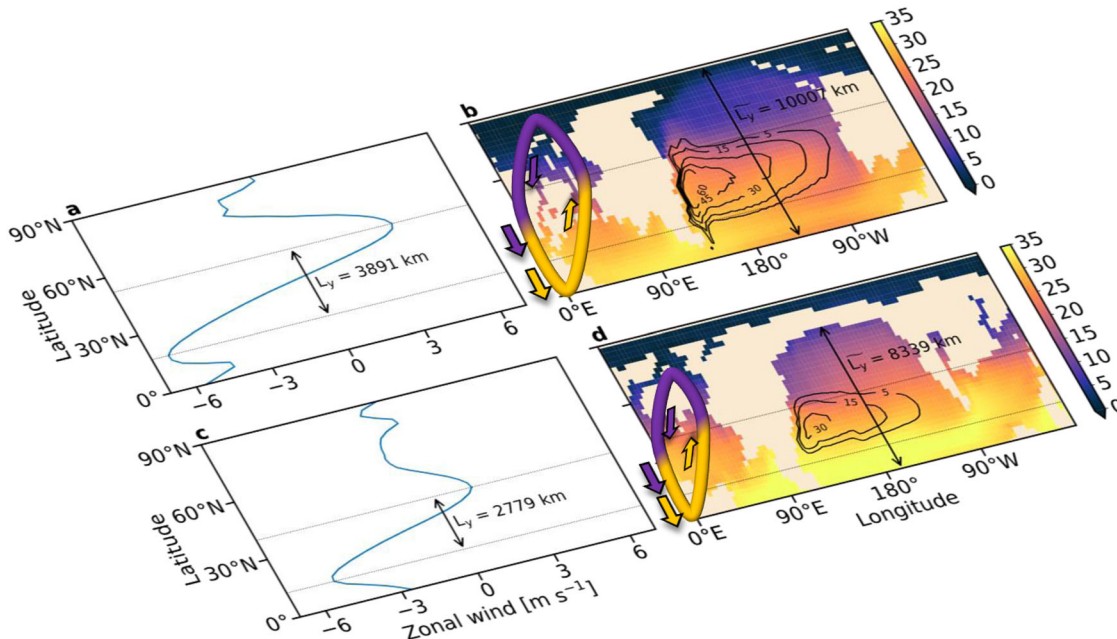

**Fig. 3 | Zonal-mean wind speed and temperature distribution in the northern paleo-Pacific Ocean for an atmospheric CO₂ concentration of 1120 ppmv in the Valanginian (~142 Ma) and Maastrichtian (~68 Ma) ages.** Panels **a** and **c** show the zonally averaged winds in the Valanginian and the Maastrichtian, respectively; the dotted lines denote the latitudes where the wind-stress curl vanishes and $L_y$ is the meridional distance between them. Panels **b** and **d** show the temperature fields (°C, color) and streamlines (Sv, contours) of the surface gyre during the two ages; $\tilde{L}_y$ is the typical Equator-to-pole extent of the basin. The yellow-purple loops depict the meridional extent of the atmospheric meridional overturning circulations. The latitude where the winds change from Easterlies to Westerlies marks the poleward extent of the atmospheric overturning cell[40].

estimated atmospheric CO₂ levels[23,35] in general and, thus by extension, in the magnitude of $\Delta SST_y$ variation we attribute to $p$CO₂ fluctuations during the period. In addition to changes in atmospheric CO₂ concentration and paleogeography, $\Delta SST_y$ can also be non-trivially affected by several other drivers in the climate system, e.g., fluctuations in deep-sea temperatures[22,36], opening/closing/deepening of oceanic gateways[10,11,37] and variations in the extent of marine ice cover[8].

The temporal variation in meridional temperature gradients obtained from the HadCM3L model for both the simulation ensembles agrees with the proxy-inferred increase in temperature gradient from the Early to the Late Cretaceous[1]. Since the atmospheric CO₂ and continental arrangement do not coevolve in the current set of simulations, we stop short of comparing the model temperature gradients with proxy data. This should be addressed in future work using data from multiple climate models in which atmospheric CO₂ and geography are varied simultaneously. Additionally, numerical simulations that use a hierarchy of coupled atmosphere–ocean models with idealized continental configurations can further advance our understanding of how paleogeography affects $\Delta SST_y$. Nevertheless, our analysis is based on foundational models of atmospheric and oceanic circulation and first-order assumptions to quantify the characteristics of a given paleoclimate — which are simple and robust. It may, therefore, also explain the effects of changes in paleogeography on variations in $\Delta SST_y$ in other geologic periods.

## Methods

### Identifying the meridional and zonal extents of the mid-latitudinal gyral basin in the North Pacific

To determine the typical meridional and zonal extents of the gyral basin in the mid-latitudinal North Pacific, we construct a 'trapezoidal' domain in the ocean. The meridional margins of the gyral basin are given by the latitudes, $\phi_1$ and $\phi_2$, where the zonally averaged wind calculated by the HadCM3L model reaches an extremum. These extrema correspond to the points where the curl of the zonally averaged wind stress vanishes. Subsequently, we determine four typical

longitudes, $\lambda_1$–$\lambda_4$, based on the continental boundaries between the latitudes $\phi_1$ and $\phi_2$ such that the region enclosed between the vertices P ($\phi_1, \lambda_1$), Q ($\phi_1, \lambda_2$), R ($\phi_2, \lambda_3$) and S ($\phi_2, \lambda_4$) denotes the 'gyral basin'. The meridional extent of the gyral basin ($L_y$) is defined as the latitudinal distance between the edges PQ and RS and the zonal extent ($L_x$) is defined as the cosine(latitude) weighted average of the lengths of PQ and RS. Supplementary Fig. 1 illustrates the gyral basins during different geologic ages in the Cretaceous and the Paleogene. We note that both $L_y$ and $L_x$ depend on the grid resolution. Despite the rather stringent definition of $L_y$ and $L_x$ and their dependence on the grid resolution, the correlation between $L_y$ and $\Delta SST_y$ is very strong, which further demonstrates the robustness of our analysis.

### Relation between meridional sea surface temperature gradients and the volume (mass) transport in the ocean

Poleward volume (mass) and heat transport associated with the atmospheric meridional overturning circulation and the gyral circulation in the ocean's surface lower the meridional sea surface temperature gradient ($\Delta SST_y$), which is driven by differential solar heating. We assume that $\Delta SST_y$ is given by (see assumption 1, main text):

$$\Delta SST_y = \Delta SST_y^{Rad}(1 - c_O \mathcal{H}_O - c_A \mathcal{H}_A) \qquad (2)$$

where $\Delta SST_y^{Rad}$ is the meridional gradient of SST at radiative equilibrium when ocean and atmosphere heat transport are zero ($\mathcal{H}_O = \mathcal{H}_A = 0$), and $c_O$ and $c_A$ are the appropriate empirical constants. Assuming that atmosphere and ocean heat transport are proportional to each other (on average, and not at every latitude)[33], we may write $c_A \mathcal{H}_A = (\gamma - 1)c_O \mathcal{H}_O$, where $\gamma > 1$ is an arbitrary constant (see assumption 2, main text). Equation (2) can then be rewritten as:

$$\Delta SST_y = \Delta SST_y^{Rad}(1 - \gamma c_O \mathcal{H}_O). \qquad (3)$$

We further assume that the gyral heat transport can be written as $\mathcal{H}_O = c_p \rho \psi_{max} \Delta SST_x$, where $\Delta SST_x$ is the East-West SST gradient[38],

yielding:

$$\Delta SST_y = \Delta SST_y^{Rad}\left(1 - \gamma c_O c_p \rho \psi_{\max} \Delta SST_x\right). \qquad (4)$$

Given the well-constrained geometry of a Stommel-like gyre, we substitute $\Delta SST_x = c_{xy}\Delta SST_y$ (see assumption 3, main text) to obtain:

$$\Delta SST_y = \frac{\Delta SST_y^{Rad}}{1 + \left(\gamma c_O c_p \rho \frac{1}{c_{xy}} \psi_{\max} \Delta SST_y^{Rad}\right)}, \qquad (5)$$

where $0 < c_{xy} \leq 1$ is another constant. Substituting $\kappa = \gamma c_O c_p \rho \frac{1}{c_{xy}}$, yields Eq. (1).

## Coupled atmosphere–ocean model description

The data analyzed in this study was generated from two ensembles of simulations carried out using the UK Met Office coupled atmosphere-ocean model, HadCM3L, which includes multiple climate feedbacks, including vegetation feedbacks[6,21]. The model has a horizontal resolution of 3.75° in longitude × 2.5° in latitude. The atmosphere and the ocean are divided into 19 and 20 vertical levels, respectively. The $CO_2$ in the model atmosphere is kept fixed at 560 ppmv and 1120 ppmv for the two ensembles of simulations, while the paleogeography and solar constant are varied. The chosen range of atmospheric $CO_2$ captures the maximum variation (530 ppmv to 840 ppmv) in the long-term trend of LOESS smoothened data on estimated atmospheric $CO_2$ concentration during the CPE[22]. The geographies corresponding to different geological ages are developed by Getech Plc using methods described in ref. 39, and are constrained by geological data aggregated from lithological, tectonic, fossil, and deep-sea studies. The geographies were originally produced at a resolution of 0.5° in both latitude and longitude, which were subsequently used to generate geographies at model resolution. Additionally, some smoothening was applied to ensure model stability. A more detailed description is available in Section 2.1 of ref. 6. Given that the variation in solar constant was quite small (a monotonic increase of ~0.9% from the Berriasian age in the Cretaceous to the Priabonian age in the Paleogene), we assume that for a given $p$CO$_2$ in the model atmosphere, any changes in the spatial distribution of SSTs between simulations arise primarily from the changing paleogeography during the CPE and the atmosphere–ocean feedbacks associated with it.

## Data availability

The simulations used in this study are identical to those described in ref. 7, and the full details are available therein. The data from the model simulations are available from https://www.paleo.bristol.ac.uk/ummodel/scripts/papers/Farnsworth_et_al_2019.html.

## Code availability

The data generated from the model simulations was analyzed using standard Python packages.

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

## Acknowledgements

K.G. and H.G. were supported by the Joint National Natural Science Foundation of China–Israel Science Foundation (research grant number 2547/17) and by the U.S.–Israel Binational Science Foundation (BSF grant number 2018152). D.J.L. and A.F. acknowledge Leverhulme grant RPG-2019-365; D.J.L. acknowledges NERC grant NE/X000222/1; A.F. acknowledges the Chinese Academy of Sciences Visiting Professorship for Senior International Scientists grant 2021FSE0001.

## Author contributions

K.G. conceived the initial idea, developed the analytical model, and applied it to the HadCM3L data. O.A., H.G., and N.P. contributed to the development of the analytical model, and S.M. contributed to the application of the analytical model to the HadCM3L data. A.F. and D.J.L. carried out the HadCM3L paleoclimate simulations. K.G. drafted the initial version of the paper, and all other authors contributed to the text.

## Competing interests

The authors declare no competing interests.
