## [Peer Review File · Nature Communications]

Effects of paleogeographic changes and CO₂ variability on northern mid-latitude temperature gradients in the CretaceousReviewer #1 (Remarks to the Author):

Review of "Both tectonic activity and CO₂ variability affect temperature gradients in the Cretaceous" by Gianchandani et al.

This paper uses a suite of climate model simulations to investigate the Cretaceous greenhouse period. In particular, the authors study modelled mid-latitude sea-surface temperature (SST) gradients in the Northern hemisphere and the respective roles of changes in continental configuration, atmospheric carbon dioxide (CO₂), and meridional heat transport in the wind-driven ocean gyres. The novel aspect of this work consists of an analysis of oceanic heat transport relating to the changing aspect ratio of the paleo-Pacific ocean basin.

This is certainly an interesting paper which will be useful to the paleoclimate community. My main recommendation would be not to focus on the Hadley model alone, but to include existing (and publicly available) ensembles of paleoclimate model simulations, in particular Li et al. (2022, Scientific Data) covering the Phanerozoic. There is also a set of Mesozoic simulations published by Landwehrs et al. (2021) using an intermediate-complexity model to disentangle the effects of paleogeography and CO₂ which is relevant in the context of this work. Finally (and just a very minor issue), the end of the Cretaceous is now usually set at 66 Ma rather than 65 Ma.

Reviewer #2 (Remarks to the Author):

Summary: Gianchandani et al. use an ensemble of HadCM3L simulations to investigate and quantify the relative contributions of atmospheric CO₂ and tectonic activity to variations in mid-latitude SSTs in the North Pacific Basin during the Cretaceous. The simulations suggest that changes in CO₂ and paleogeography had comparable impacts on mid-latitude SST gradients. Gianchandani et al. also highlight the importance of ocean basin aspect ratio in determining gyral circulation, heat transport, and mid-latitude SST patterns in the North Pacific Basin by developing a fundamental model of wind-driven ocean gyres.

- What are the noteworthy results?

The main finding of this study is that changes in CO₂ and paleogeography contribute similarly to increasing mid-latitude SST gradients over the Cretaceous. Although it is well established in the literature that changes in CO₂ and paleogeography contributed to SST variability over the Cretaceous, this finding is noteworthy because the relative contributions of CO₂ and tectonic activity to SST variability are not established. Another noteworthy result is that a decrease in gyral volume transport, and by extension ocean basin aspect ratio, can account for ~80% variability in the meridional SST gradient in the mid-latitude North Pacific over the Cretaceous.

- Will the work be of significance to the field and related fields? How does it compare to the established literature? If the work is not original, please provide relevant references.

Yes, this work is significant for the understanding of ocean circulation and temperature during the Cretaceous. The study outlines a simple model of wind-driven gyres based on the horizontal aspect ratio of the ocean basin that may be applied to other deep-time intervals to better understand the contribution of tectonic activity to changes in ocean circulation and temperature.

- Does the work support the conclusions and claims, or is additional evidence needed?

Overall, yes. However, my main concern is that the range of CO₂ captured in these simulations does not capture the inferred paleo-CO₂ range of the Cretaceous based on Foster et al. (2017), which could have reached decreased to values of ~280 ppmv during the Late Cretaceous (Lines 116-120). The justification for the CO₂ levels of the simulations needs to be explained, particularly because the relationship between CO₂ and meridional SST gradient supported by the simulations (at 560-1120 ppmv) is being extended beyond this range of CO₂ (down to ~280 ppmv). This

directly impacts the how the relationship between CO₂ and meridional SST gradient is calculated for the Cretaceous, which is one of the main findings of the study.

- Are there any flaws in the data analysis, interpretation, and conclusions? Do these prohibit publication or require revision?

The statement that atmospheric CO₂ fluctuations and paleogeographic changes have a comparable effect on mid-latitude SST gradient through the Cretaceous may be too strong of a statement given that the study is restricted to the North Pacific Basin. The authors state that the fundamental model of wind-driven ocean gyres does not apply to the Southern Hemisphere (Lines 193-197). Some revisions would help broaden the conclusions of the study by investigating the relative importance of CO₂ and tectonic activity on SST variability in other ocean basins during the Cretaceous and/or demonstrating how the fundamental model of gyral circulation could also be applied to other time intervals.

- Is the methodology sound? Does the work meet the expected standards in your field?

Overall, yes. However, the methodology requires more justification and explanation CO₂ and paleogeographic reconstructions used for the simulations. Both CO₂ level and paleogeographic reconstruction are subject to uncertainty that should be more explicitly described.

- Is there enough detail provided in the methods for the work to be reproduced? Yes.

Major comments:

- The title reiterates knowledge that is already well established about Cretaceous SSTs and does not highlight the novelty or impact of the study. This title could instead highlight the relative importance of CO₂/tectonic activity in the northern mid-latitudes

- Main/Introduction: The gap in knowledge motivating this study and broader scientific context from previous literature needs to be explained in the introduction. The first paragraph is already diving into methodology of the study, which has only been motivated by a few sentences. As a result, I am unsure if this study is motivated primarily by a desire to better understand SST variability during the Cretaceous or develop a more general framework for understanding SST variability based on gyral circulation/basin geometry in which the Cretaceous is a case study. After the sentence that motivates looking at other environmental factors (beyond just CO₂) (Lines 38-41), there should be a brief discussion of the work that has investigated the impacts of CO₂ and paleogeography on ocean temperature and circulation during the Cretaceous to explicitly outline the novelty of the current study (e.g., Ladant et al, 2020 in *Climate of the Past*; Donnadieu et al., 2016 in *Nature Comm*).

- Lines 44-45: Why did you choose 560 and 1120 ppmv CO₂? This should be justified in the methods section. Although 560 and 1120 ppmv capture most of the inferred paleo-CO₂ variability during the Cretaceous, CO₂ levels could have been much lower than 560 ppmv during the Late Cretaceous, as you state on Lines 116-120. Regarding Lines 116-120, why do you expect that the relationship between meridional SST gradient and CO₂ between two relatively high CO₂ levels (560 and 1120 ppmv) would extend to lower CO₂?

- Lines 177-179: Although the simple model explains a large proportion of the variance (Fig. 2A), it shows a roughly linear trend of increasing SST gradient at both CO₂ concentrations, and thus does not capture a threshold-like behavior due to paleogeographic changes between ~130-110 Mya, which is very interesting. Before and after this interval, the changes in SST gradient are much smaller. What changes/feedbacks associated with those paleogeographic changes are creating such a dramatic increase in SST gradient during that specific interval?

Minor comments:

- Figure 1: I would suggest making red/blue lines and markers thicker to differentiate from black arrows and making black contour labels larger/more visible (I cannot read them)

- Line 189-191: Two "results" in the sentence, perhaps break up this sentence with commas or choose a different combination of words here.
- Line 197-200: Citations from previous work should be added here.
- Line 206: Add period to end of sentence.

Reviewer #3 (Remarks to the Author):

Please find my comments in the attached word file.

Reviewer #3 Attachment on the following page

Review: „Both tectonic activity and CO₂ variability affect temperature gradients in the Cretaceous”

Kaushal Gianchandani, Sagi Maor, Ori Adam, Alexander Farnsworth, Hezi Gildor, Daniel J. Lunt, Nathan Paldor

Gianchandani et al. present a study that uses general circulation modeling to investigate the driving mechanisms of changes in the Cretaceous meridional temperature. The authors conclude that both paleogeographic changes and decreasing $p\text{CO}_2$ contributed equally to a decrease in the midlatitude temperature gradient of about 6 °C from the Early to Late Cretaceous (i.e., from the Valanginian to the Maastrichtian). This change in temperature gradient was caused by a decrease in oceanic heat transport resulting from a weakening of the wind-driven gyre circulation in the northern Paleo-Pacific in response to a decreased latitudinal extent of the Pacific basin.

As someone who has worked a lot on Cretaceous paleoceanography and paleoclimatology, but is not an expert in general circulation modeling, I find the manuscript to be well written, well structured, and easy to follow. The science is sound and the interpretations are supported by the data. The conclusions are presented in a clear manner and are relevant to the broader geoscientific community. As such, I consider the manuscript to be well suited for publication in Nature Communications and have only a few minor comments.

- 1.) Paleogeography: I recommend including some background information on the paleogeography implemented in the model in the Methods section. Because paleogeographic restorations of the Paleo-Pacific are complicated by extensive subduction of oceanic crust since the Cretaceous, it would be particularly helpful to include a brief comparison between the implemented paleogeography and other published plate tectonic models to show that the general trends in L_x and L_y are consistent across the various available models.
- 2.) Role of sea-level change: Do the changes in land-sea distribution shown in Fig. S1 and Fig. S6 reflect solely plate tectonic processes (i.e., changes in the distribution of oceanic and continental crust) or also global sea-level changes (i.e., flooding of continents)? If both processes are involved, the authors may consider replacing "tectonic activity" with "paleogeographic changes" in the title of the manuscript.
- 3.) Paleo-SST data: I suggest to include a graphical comparison of proxy-based SST estimates and model-derived latitudinal temperature gradients, at least for the Valanginian and Maastrichtian, to demonstrate that the model is capable of capturing the general patterns evident in the proxy data.
- 4.) Figure 1: The labeling of the contour lines in Figures 1A and 1B is barely visible.
- 5.) Figure 3: Correct "Maastrichtian" in the caption of the Figure.
- 6.) Line 95: Please indicate whether ΔSST were averaged over the Pacific basin or globally.
- 7.) Line 192: Please provide add a short sentence or a reference to explain what tectonic processes drove the changes in the northern hemisphere paleogeography, specifically the closure of the "proto-Bering Strait" connecting the northernmost Pacific and the Arctic Ocean.

Re: Point-by-point response to reviewers' comments

Reviewer #1:

This paper uses a suite of climate model simulations to investigate the Cretaceous greenhouse period. In particular, the authors study modelled mid-latitude sea-surface temperature (SST) gradients in the Northern hemisphere and the respective roles of changes in continental configuration, atmospheric carbon dioxide (CO₂), and meridional heat transport in the wind-driven ocean gyres. The novel aspect of this work consists of an analysis of oceanic heat transport relating to the changing aspect ratio of the paleo-Pacific ocean basin.

We thank the reviewer for her/his positive remarks on our manuscript and appreciate the suggestion to analyze the datasets provided by Li et al. (2022) and Landwehrs et al. (2021). We summarize the findings from our examination of the two datasets below.

This is certainly an interesting paper which will be useful to the paleoclimate community. My main recommendation would be not to focus on the Hadley model alone, but to include existing (and publicly available) ensembles of paleoclimate model simulations, in particular Li et al. (2022, Scientific Data) covering the Phanerozoic. There is also a set of Mesozoic simulations published by Landwehrs et al. (2021) using an intermediate-complexity model to disentangle the effects of paleogeography and CO₂ which is relevant in the context of this work. Finally (and just a very minor issue), the end of the Cretaceous is now usually set at 66 Ma rather than 65 Ma.

The high-resolution data provided in Li et al. (2022) does not contain the ocean variables relevant to our analysis, in particular, the sea surface temperature (SST) and the volumetric (mass) transport corresponding to the surface layer (stream function, Ψ). Moreover, in the simulations discussed in Li et al. (2022), both the atmospheric CO₂ concentration and the paleogeography change simultaneously, which makes it difficult to disentangle the respective contribution of each of these components to the changes in the climate.

The ensemble of simulations discussed in Landwehrs et al. (2021) shows a decrease in mid-litudinal SST gradient during the Cretaceous period, which is in contrast with what the paleotemperature proxies suggest (O'Brien et al., 2017). Furthermore, the fast statistical-dynamical atmospheric model in CLIMBER-3 α employed for these simulations has a rather coarse resolution of 7.5° in latitude (compared with 2.5° in the HadCM3L model). Thus, it does not capture the subtle variations in the curl of the surface wind-stress induced by paleogeographic changes and the corresponding changes in the distance between wind-stress extrema (L_y) on geological timescales, which underpin our analysis.

Given these constraints, we are unable to extend our analysis to other ensembles of paleo-climate simulations at this stage. Nonetheless, we are grateful for the reviewer's suggestion.

We have modified the end of the Cretaceous from 66 Ma to 65 Ma as per the reviewer's suggestion, thank you.

Reviewer #2 (Remarks to the Author):

Summary: Gianchandani et al. use an ensemble of HadCM3L simulations to investigate and quantify the relative contributions of atmospheric CO₂ and tectonic activity to variations in mid-latitude SSTs in the North Pacific Basin during the Cretaceous. The simulations suggest that changes in CO₂ and paleogeography had comparable impacts on mid-latitude SST gradients. Gianchandani et al. also highlight the importance of ocean basin aspect ratio in determining gyral circulation, heat transport, and mid-latitude SST patterns in the North Pacific Basin by developing a fundamental model of wind-driven ocean gyres.

• What are the noteworthy results?

The main finding of this study is that changes in CO₂ and paleogeography contribute similarly to increasing mid-latitude SST gradients over the Cretaceous. Although it is well established in the literature that changes in CO₂ and paleogeography contributed to SST variability over the Cretaceous, this finding is noteworthy because the relative contributions of CO₂ and tectonic activity to SST variability are not established. Another noteworthy result is that a decrease in gyral volume transport, and by extension ocean basin aspect ratio, can account for ~80% variability in the meridional SST gradient in the mid-latitude North Pacific over the Cretaceous.

• Will the work be of significance to the field and related fields? How does it compare to the established literature? If the work is not original, please provide relevant references.

Yes, this work is significant for the understanding of ocean circulation and temperature during the Cretaceous. The study outlines a simple model of wind-driven gyres based on the horizontal aspect ratio of the ocean basin that may be applied to other deep-time intervals to better understand the contribution of tectonic activity to changes in ocean circulation and temperature.

• Does the work support the conclusions and claims, or is additional evidence needed?

Overall, yes. However, my main concern is that the range of CO₂ captured in these simulations does not capture the inferred paleo-CO₂ range of the Cretaceous based on Foster et al. (2017), which could have reached decreased to values of ~280 ppmv during the Late Cretaceous (Lines 116-120). The justification for the CO₂ levels of the simulations needs to be explained, particularly because the relationship between CO₂ and meridional SST gradient supported by the simulations (at 560-1120 ppmv) is being extended beyond this range of CO₂ (down to ~280 ppmv). This directly impacts the how the relationship between CO₂ and meridional SST gradient is calculated for the Cretaceous, which is one of the main findings of the study.

• Are there any flaws in the data analysis, interpretation, and conclusions? Do these prohibit publication or require revision?

The statement that atmospheric CO₂ fluctuations and paleogeographic changes have a comparable effect on mid-latitude SST gradient through the Cretaceous may be too strong of a statement given that the study is restricted to the North Pacific Basin. The authors state that the fundamental model of wind-driven ocean gyres does not apply to the Southern Hemisphere (Lines 193-197). Some revisions would help broaden the conclusions of the study by investigating the relative importance of CO₂ and tectonic activity on SST variability in other ocean basins during the Cretaceous and/or demonstrating how the fundamental model of gyral circulation could also be applied to other time intervals.

• Is the methodology sound? Does the work meet the expected standards in your field?

Overall, yes. However, the methodology requires more justification and explanation of CO₂ and paleogeographic reconstructions used for the simulations. Both CO₂ level and paleogeographic reconstruction are subject to uncertainty that should be more explicitly described.

• Is there enough detail provided in the methods for the work to be reproduced?
Yes.

We thank the reviewer for the overall positive comments and for providing us some critical feedback which has helped us improve the manuscript. We provide a point-by-point response to her/his comments below.

Major comments:

• The title reiterates knowledge that is already well established about Cretaceous SSTs and does not highlight the novelty or impact of the study. This title could

instead highlight the relative importance of CO₂/tectonic activity in the northern mid-latitudes

We have changed the manuscript's title to "Effects of paleogeographic changes and CO₂ variability on northern mid-latitude temperature gradients in the Cretaceous".

• Main/Introduction: The gap in knowledge motivating this study and broader scientific context from previous literature needs to be explained in the introduction. The first paragraph is already diving into methodology of the study, which has only been motivated by a few sentences. As a result, I am unsure if this study is motivated primarily by a desire to better understand SST variability during the Cretaceous or develop a more general framework for understanding SST variability based on gyral circulation/basin geometry in which the Cretaceous is a case study. After the sentence that motivates looking at other environmental factors (beyond just CO₂) (Lines 38-41), there should be a brief discussion of the work that has investigated the impacts of CO₂ and paleogeography on ocean temperature and circulation during the Cretaceous to explicitly outline the novelty of the current study (e.g., Ladant et al, 2020 in *Climate of the Past*; Donnadieu et al., 2016 in *Nature Comm*).

The focus of our study is to better understand how the interplay between fluctuations in atmospheric CO₂ and paleogeography driven changes in the ocean's circulation affected the meridional SST gradients during the Cretaceous. Given that surface ocean gyres are a prominent feature of the ocean's circulation across geological timescales, we develop a general framework to quantify the effect of paleogeography driven changes in surface ocean circulation which is applicable to well constrained Stommel like ocean basins which persist in climate model simulations of multiple geologic periods including the Cretaceous.

We thank the reviewer for suggesting that we clarify this further. We have incorporated multiple relevant references regarding how paleogeographic alterations during the Cretaceous affect the climate and the ocean's circulation during the period (Lines 45-52). Additionally, we explicitly state the novelty of our study in Lines 64-70 and specify our aim in Lines 90-93. Furthermore, we have extended our analysis to include 7 ages from the Paleogene period as well to show the applicability of our model to geologic times other than the Cretaceous (see Fig. 2 and Lines 225-229).

• Lines 44-45: Why did you choose 560 and 1120 ppmv CO₂? This should be justified in the methods section. Although 560 and 1120 ppmv capture most of the inferred paleo-CO₂ variability during the Cretaceous, CO₂ levels could have been much lower than 560 ppmv during the Late Cretaceous, as you state on Lines 116-120. Regarding Lines 116-120, why do you expect that the relationship between meridional SST gradient and CO₂ between two relatively high CO₂ levels (560 and 1120 ppmv) would extend to lower CO₂?

We elaborate on our choice of CO₂ in Lines 96 – 105 of the revised manuscript. Further, we now focus only on the implications that variation in the long-term trend in estimated atmospheric CO₂ levels have on the temperature gradient (Lines 163 – 180). Our chosen range of atmospheric CO₂ in the simulations captures the maximum variation in the long-term trend in estimated atmospheric CO₂ during the Cretaceous period (Valdes et al., 2021) and this is discussed in the revised manuscript as well (Lines 96 – 102). We also acknowledge that there is an uncertainty in the estimated atmospheric CO₂ and by extension in the magnitude of variation in temperature gradient we attribute to fluctuation in atmospheric CO₂ during the Cretaceous (Lines 270 – 274). Note also that the apparently low values of CO₂ as inferred solely from CO₂ proxies in Foster et al (2017) are inconsistent with relatively warm temperatures at this time inferred from multiple SST proxies (no less than 20 °C global annual mean; Scotese et al., 2021)

• **Lines 177-179: Although the simple model explains a large proportion of the variance (Fig. 2A), it shows a roughly linear trend of increasing SST gradient at both CO₂ concentrations, and thus does not capture a threshold-like behavior due to paleogeographic changes between ~130-110 Mya, which is very interesting. Before and after this interval, the changes in SST gradient are much smaller. What changes/feedbacks associated with those paleogeographic changes are creating such a dramatic increase in SST gradient during that specific interval?**

We agree that the steep increase in temperature gradients between 130 and 100 Ma is an interesting feature. In the revised version, we highlight this and acknowledge that it lies outside the range in which our best-fit curve is expected to vary. The steep increase stems from the abrupt decrease in SST at 50°N (Fig. S5, Revised Manuscript). This can potentially be related to vanishing of the sub-polar gyre from the Barremian to the Albian and the entrainment of high-latitude cold water by the mid-latitude gyre (Fig. R1A-C). In the present-day Atlantic, there is clear demarcation between the mid-latitude and the subpolar gyres, which limits the mixing of water at the poleward edge (Equatorward edge) of the mid-latitude (subpolar) gyre. The vanishing of the subpolar gyres from the Barremian to the Albian age could lead to 'blurring' of this demarcation. The northern branch of mid-latitude gyre could then entrain some of the cold high-latitude water, which can potentially explain the abrupt cooling from the Barremian to the Albian age.

However, other factors including paleogeography driven polar amplification could also contribute to the cooling effect. Examining the precise climatic variations/feedbacks associated with paleogeographic changes that lead to this abrupt cooling is beyond the scope of the present short manuscript. This is discussed in Lines 240 - 248.

Figure R1 | The stream function (A-C) and the temperature fields (D-F) corresponding to the surface layer for three geologic ages in the Cretaceous. A,D depict the Albian age (~106 Ma), B,E depict the Aptian age (~119 Ma) and C,F depict the Barremian age (~128 Ma). The solid (dashed) contours in A-C depict the streamlines of constant anti-cyclonic (cyclonic) mass transport. The dotted lines in all six panels mark the 20°N and 50°N latitudes.

Minor comments:

- **Figure 1: I would suggest making red/blue lines and markers thicker to differentiate from black arrows and making black contour labels larger/more visible (I cannot read them)**

The suggestion was incorporated and the figure was modified accordingly.

- **Line 189-191: Two “results” in the sentence, perhaps break up this sentence with commas or choose a different combination of words here.**

The sentence was modified to present the results more clearly, see Lines 260 – 264.

- **Line 197-200: Citations from previous work should be added here.**

Included, see Lines 276 – 278.

- **Line 206: Add period to end of sentence.**

Added.

Reviewer #3:

Gianchandani et al. present a study that uses general circulation modeling to investigate the driving mechanisms of changes in the Cretaceous meridional temperature. The authors conclude that both paleogeographic changes and decreasing pCO₂ contributed equally to a decrease in the midlatitude temperature gradient of about 6 °C from the Early to Late Cretaceous (i.e., from the Valanginian to the Maastrichtian). This change in temperature gradient was caused by a decrease in oceanic heat transport resulting from a weakening of the wind-driven gyre circulation in the northern Paleo-Pacific in response to a decreased latitudinal extent of the Pacific basin. As someone who has worked a lot on Cretaceous paleoceanography and paleoclimatology, but is not an expert in general circulation modeling, I find the manuscript to be well written, well structured, and easy to follow. The science is sound and the interpretations are supported by the data. The conclusions are presented in a clear manner and are relevant to the broader geoscientific community. As such, I consider the manuscript to be well suited for publication in Nature Communications and have only a few minor comments.

We thank the reviewer for the encouraging remarks and for providing us critical feedback which has helped us improve the manuscript. We provide a point-by-point response below.

1.) Paleogeography: I recommend including some background information on the paleogeography implemented in the model in the Methods section. Because paleogeographic restorations of the Paleo-Pacific are complicated by extensive subduction of oceanic crust since the Cretaceous, it would be particularly helpful to include a brief comparison between the implemented paleogeography and other published plate tectonic models to show that the general trends in L_x and L_y are consistent across the various available models.

In accordance with the reviewer's suggestion, we have further elaborated on the paleogeography implemented in the HadCM3L model in the Data Availability section of the revised manuscript (Lines 441-449). We agree that a comparison between L_x and L_y as obtained from multiple models will strengthen our sermon and we attempt to do the same by examining the publicly available data provided by Li et al. (2022) and Landwehrs et al. (2021). However, we are unable to present a comparison at this stage for the following reasons:

- a) In the simulations discussed in Li et al. (2022), both the atmospheric CO₂ concentration and the paleogeography change simultaneously which makes it difficult to disentangle the respective contribution of each of these components to the changes in the curl of the surface wind-stress and the corresponding changes in the distance between wind-stress extrema (L_y).

- b) The atmospheric component of the climate model (CLIMBER-3 α) employed for the simulations discussed in Landwehrs et al. (2021) has a rather coarse resolution of 7.5° in latitude. Thus, it does not resolve the paleogeography driven changes in the meridional extent of the Hadley cell or the subtle variations in the surface winds which is critical for calculating the changes in the gyral circulation and therefore the oceanic heat transport.

We note that our approach for quantifying the effect of aspect ratio $\left(\frac{L_y}{L_x}\right)$ on the intensity of gyral circulation was applied in a previous study to the present-day ocean gyres in different ocean basins (Gianchandani et al., 2021). We showed that the small volumetric (mass) transport of the East Australian Current compared to other Western Boundary Currents can be attributed to the geometry of the South Pacific Basin, i.e., Australia's Eastern coastline is not long enough to support a strong WBC in the zonally wide South Pacific Ocean.

2.) Role of sea-level change: Do the changes in land-sea distribution shown in Fig. S1 and Fig. S6 reflect solely plate tectonic processes (i.e., changes in the distribution of oceanic and continental crust) or also global sea-level changes (i.e., flooding of continents)? If both processes are involved, the authors may consider replacing "tectonic activity" with "paleogeographic changes" in the title of the manuscript.

The change in land-sea distribution does include a sea level component and in general our chosen paleogeographic represent the high-stands in sea level. Thus, we thank the reviewer for pointing this out accept the suggestion to replace tectonic activity with paleogeographic changes in the title and. The modified title now is:

“Effects of paleogeographic changes and CO₂ variability on northern mid-latitudinal temperature gradients in the Cretaceous.”

3.) Paleo-SST data: I suggest to include a graphical comparison of proxy-based SST estimates and model-derived latitudinal temperature gradients, at least for the Valanginian and Maastrichtian, to demonstrate that the model is capable of capturing the general patterns evident in the proxy data.

We do not compare the model-derived temperature gradients with proxy data since the atmospheric CO₂ and continental arrangement do not coevolve in the current set of simulations. These simulations are not designed to represent actual changes in climate, but instead are idealized with constant CO₂ concentrations through time. However, we agree with the reviewer's suggestion that it is important to benchmark model data against geochemical proxies and plan to address it in a future work using data from multiple climate models in which atmospheric CO₂ and geography are varied simultaneously (Lines 279-286).

4.) Figure 1: The labeling of the contour lines in Figures 1A and 1B is barely visible.

The suggestion was incorporated and the figure was modified accordingly.

5.) Figure 3: Correct “Maastrichtian” in the caption of the Figure.

Corrected.

6.) Line 95: Please indicate whether Δ SST were averaged over the Pacific basin or globally.

Specified.

7.) Line 192: Please provide add a short sentence or a reference to explain what tectonic processes drove the changes in the northern hemisphere paleogeography, specifically the closure of the “proto-Bering Strait” connecting the northernmost Pacific and the Arctic Ocean.

Relevant references added.

References:

- Gianchandani, K., Gildor, H., & Paldor, N. (2021). On the role of domain aspect ratio in the westward intensification of wind-driven surface ocean circulation. *Ocean Science*, 17(1), 351–363. <https://doi.org/10.5194/os-17-351-2021>
- Landwehrs, J., Feulner, G., Petri, S., Sames, B., & Wagneich, M. (2021). Investigating Mesozoic Climate Trends and Sensitivities With a Large Ensemble of Climate Model Simulations. *Paleoceanography and Paleoclimatology*, 36(6), e2020PA004134. <https://doi.org/https://doi.org/10.1029/2020PA004134>
- Li, X., Hu, Y., Guo, J., Lan, J., Lin, Q., Bao, X., Yuan, S., Wei, M., Li, Z., Man, K., Yin, Z., Han, J., Zhang, J., Zhu, C., Zhao, Z., Liu, Y., Yang, J., & Nie, J. (2022). A high-resolution climate simulation dataset for the past 540 million years. *Scientific Data*, 9(1), 371. <https://doi.org/10.1038/s41597-022-01490-4>
- O'Brien, C. L., Robinson, S. A., Pancost, R. D., Sinninghe Damsté, J. S., Schouten, S., Lunt, D. J., Alsenz, H., Bornemann, A., Bottini, C., Brassell, S. C., Farnsworth, A., Forster, A., Huber, B. T., Inglis, G. N., Jenkyns, H. C., Linnert, C., Littler, K., Markwick, P., McAnena, A., ... Wrobel, N. E. (2017). Cretaceous sea-surface temperature evolution: Constraints from TEX86 and planktonic foraminiferal oxygen isotopes. *Earth-Science Reviews*, 172(July), 224–247. <https://doi.org/10.1016/j.earscirev.2017.07.012>
- Scotese, C. R., Song, H., Mills, B. J. W., & van der Meer, D. G. (2021). Phanerozoic paleotemperatures: The earth's changing climate during the last 540 million years.

Earth-Science Reviews, 215, 103503.
<https://doi.org/10.1016/J.EARSCIREV.2021.103503>

Valdes, P. J., Scotese, C. R., & Lunt, D. J. (2021). Deep ocean temperatures through time. *Climate of the Past*, 17(4), 1483–1506. <https://doi.org/10.5194/cp-17-1483-2021>

Reviewer #1 (Remarks to the Author):

After reading the revised manuscript as well as the point-by-point response to the reviewers' comments, I have only a few very minor comments:

Lines 17, 38, 227 (twice): In contrast to the statement in the authors' response, the age for the end of the Cretaceous has in fact not been corrected from 65 Ma to 66 Ma during revision... ;)

Caption Figure 3: Please correct "~65 Ma" to "~68 Ma".

Line 138: I am aware that this change is due to a suggestion of Reviewer 3, but "global zonally averaged SST" sounds weird. Maybe rather specify that the average was computed "over all longitudes" or similar?

Reviewer #2 (Remarks to the Author):

I appreciate the work that the Gianchandani et al. have accomplished to better motivate and expand the impact of the study. Specifically, the extension of the gyral circulation model from the Cretaceous to the Paleogene is a great addition. However, much of the manuscript (title, abstract, introduction, methods, some results, and discussion) have not all been equally updated to reflect the inclusion of the Paleogene and how this alters/broadens the impact and conclusions of the study. I have included more detailed suggestions in the comments below.

Major Comments

- Introduction: You have provided the background to explain how your study advances our understanding of Cretaceous SSTs and ocean circulation, but you have expanded the scope to include the Paleogene which is not mentioned in the introduction. Additionally, you have not fully incorporated the paleoclimate and gyre circulation components of the introduction. One suggestion to incorporate Cretaceous-Paleogene climate with your gyre model in the introduction is to introduce (1) description of meridional SST patterns in the Cretaceous-Paleogene, (2) problem: quantifying the impact of paleogeography on these SST patterns is important but complicated as previous work has shown, (3) gap: surface gyres are an important driver of SST patterns and have not been directly linked to paleogeographic changes in previous work, (4) solution: Stommel's simple model of gyral circulation can be used to quantify the influence of paleogeography on past SST patterns, and thus better understand the relative importance of other climate forcings like CO₂. I would also introduce all of these general ideas before including any details about your simulations (lines 53-56).
- The Results section, including "Wind-driven gyral circulation in the Cretaceous" should include a justification of why Stommel's model also works for the Paleogene
- The Results section, "Variability in meridional SST gradients and its dependence on atmospheric CO₂ and paleogeography", does not touch on the differences in long-term trends of meridional SST gradients during the Paleogene compared to the Cretaceous and this is a new interesting component of the study.
- Line 188: I think this should be modified to "compared to long-term trends in pCO₂". I think you must be careful with these statements because you are not capturing the entire inferred range of CO₂ during this Cretaceous-Paleogene (you are capturing the range of the smoothed long-term trend). You may even add a sentence that states that CO₂ may have a larger impact if shorter-term fluctuations in atmospheric CO₂ and/or if low CO₂ estimates from some proxies (line 102-103) are accounted for.
- Lines 253-258: How does this logic extend from the Cretaceous through the Paleogene? My understanding is that Figure S7 demonstrates that land area fraction continues to increase through the Paleocene and Eocene, but in Figure S6 there is not a concurrent decrease in Ly with a decrease geographical extent of the ocean basin during this time. The relationship between Ly and maximal volume transport with SST gradient seems to hold from the Cretaceous-Paleogene, but this does not track with the geographical extent of the ocean basin during the Paleogene. If I understand that correctly, is there something other than paleogeography that is driving an increase in Ly during the Paleogene?

- Lines 435-454: I think much of these simulation details should be moved out of the Data Availability statement into the the first paragraph of the Results "Numerical Simulations" to outline basic structure/motivation of simulations used and details should be placed in a Methods section at the end. I'm thinking of an organization like Sauermilch et al. (2021) in Nature Comms <https://doi.org/10.1038/s41467-021-26658-1>

Minor Comments

- Line 189: I would add a topic sentence to this paragraph saying that differences in long-term SST gradients between the simulations and paleotemperature reconstructions may be attributed to the limited spatial distribution of available proxies (the topic of this paragraph is unclear from the first sentence so the transition from the last paragraph to this one is difficult to understand)
- Lines 234-239: Split this into two sentences at "which"

Reviewer #3 (Remarks to the Author):

The authors have done a thorough job of addressing my concerns. I especially welcome the expanded discussion on the model's paleogeography and recommend the publication of the manuscript.

Re: Point-by-point response to reviewers' comments

The reviewers' comments are highlighted in **blue** and the authors' response is in **black**.

Reviewer #1 (Remarks to the Author):

After reading the revised manuscript as well as the point-by-point response to the reviewers' comments, I have only a few very minor comments:

1. Lines 17, 38, 227 (twice): In contrast to the statement in the authors' response, the age for the end of the Cretaceous has in fact not been corrected from 65 Ma to 66 Ma during revision... ;)

We thank the reviewer for pointing this oversight on our part, the end of the Cretaceous has now been corrected to 66 Ma throughout the manuscript.

2. Caption Figure 3: Please correct "~65 Ma" to "~68 Ma".

Done.

3. Line 138: I am aware that this change is due to a suggestion of Reviewer 3, but "global zonally averaged SST" sounds weird. Maybe rather specify that the average was computed "over all longitudes" or similar?

Done. See line 145-146 in the revised version.

Reviewer #2 (Remarks to the Author):

I appreciate the work that the Gianchandani et al. have accomplished to better motivate and expand the impact of the study. Specifically, the extension of the gyral circulation model from the Cretaceous to the Paleogene is a great addition. However, much of the manuscript (title, abstract, introduction, methods, some results, and discussion) have not all been equally updated to reflect the inclusion of the Paleogene and how this alters/broadens the impact and conclusions of the study. I have included more detailed suggestions in the comments below.

We thank the reviewer for the kind remarks about the work we have put in revising the manuscript. We have taken all the comments that s/he has made into consideration while preparing the revised version of the manuscript. These comments have definitely helped us broadening the impact of our research. Please note that while we discuss the novel findings regarding the Paleogene in the "Results" section, we refrain from modifying the title and the abstract for reasons discussed in the response to major comment # 3.

Major Comments

1. Introduction: You have provided the background to explain how your study advances our understanding of Cretaceous SSTs and ocean circulation, but you have expanded the scope to include the Paleogene which is not mentioned in the introduction. Additionally, you have not fully incorporated the paleoclimate and gyre circulation components of the introduction. One suggestion to incorporate Cretaceous-Paleogene climate with your gyre model in the introduction is to introduce (1) description of meridional SST patterns in the Cretaceous-Paleogene, (2) problem: quantifying the impact of paleogeography on these SST patterns is important but complicated as previous work has shown, (3) gap: surface gyres are an important driver of SST patterns and have not been directly linked to paleogeographic changes in previous work, (4) solution: Stommel's simple model of gyral circulation can be used to quantify the influence of paleogeography on past SST patterns, and thus better understand the relative importance of other climate forcings like CO₂. I would also introduce all of these general ideas before including any details about your simulations (lines 53-56).

We restructured the introduction along the four pointers highlighted by the reviewer as can be seen in the Lines marked by blue/red in the track changes version (Lines 33-37, 49-56, 59-61, 77-88, 90-91). The modifications also include a discussion the Paleogene SST gradients and the two new references that underscores the effect of paleogeographic changes on the gradients.

2. The Results section, including "Wind-driven gyral circulation in the Cretaceous" should include a justification of why Stommel's model also works for the Paleogene

Stommel's model works for both the Cretaceous and the Paleogene because despite considerable tectonic activity the zonal dimensions of the ocean basin remained ~13000 km for the span of 110 million years (the Cretaceous-Paleocene-Eocene, PCE, epochs). We underscore this in the revised manuscript, see Lines 119-121. Furthermore, this point is reiterated in the "Results" section where we state that >60% of the oceanic volumetric transport was in the large paleo pacific during the PCE. See Lines 244-245.

3. The Results section, "Variability in meridional SST gradients and its dependence on atmospheric CO₂ and paleogeography", does not touch on the differences in long-term trends of meridional SST gradients during the Paleogene compared to the Cretaceous and this is a new interesting component of the study.

We expanded the subsection entitled “Variability in meridional SST gradients and its dependence on atmospheric CO₂ and paleogeography”, which now includes a discussion of the Paleogene. See Lines 136-137, 139-140, 149-151, 161-162, 172-176, 189-191, 200-202. However, we still keep the discussion in the paper focused around the Cretaceous in particular since ΔSST_y variation during the Paleogene is of the order of the standard deviation around the ensemble averaged global mean SST.

4. Line 188: I think this should be modified to “compared to long-term trends in pCO₂”. I think you must be careful with these statements because you are not capturing the entire inferred range of CO₂ during this Cretaceous-Paleogene (you are capturing the range of the smoothed long-term trend). You may even add a sentence that states that CO₂ may have a larger impact if shorter-term fluctuations in atmospheric CO₂ and/or if low CO₂ estimates from some proxies (line 102-103) are accounted for.

We thank the reviewer for highlighting this subtle issue. We have now modified the text to further emphasize the sensitive dependence of our results and inferences on the choice of atmospheric CO₂ concentration. See Lines 193-196.

5. Lines 253-258: How does this logic extend from the Cretaceous through the Paleogene? My understanding is that Figure S7 demonstrates that land area fraction continues to increase through the Paleocene and Eocene, but in Figure S6 there is not a concurrent decrease in L_y with a decrease geographical extent of the ocean basin during this time. The relationship between L_y and maximal volume transport with SST gradient seems to hold from the Cretaceous-Paleogene, but this does not track with the geographical extent of the ocean basin during the Paleogene. If I understand that correctly, is there something other than paleogeography that is driving an increase in L_y during the Paleogene?

The reviewer is correct in pointing out that factors other than the Equator-to-pole extent of the ocean basin can affect L_y on geological timescales. A previous paper by Adam et al., 2022 discusses the correlations between L_y and several other climatic features such as the meridional extent/intensity of the Hadley cell, the intensity of the Walker circulation, zonal asymmetries in the SST fields. Changes in one or more of these features can potentially be a result of the opening of Atlantic in the Paleogene, which can in turn affect L_y . However, a causal link between changes in climate features and the corresponding effect on L_y has not yet been established and should be explored in a future work. We thank the reviewer for bringing this issue up and we acknowledge this in Lines 272-277.

6. Lines 435-454: I think much of these simulation details should be moved out of the Data Availability statement into the the first paragraph of the Results “Numerical Simulations” to outline basic structure/motivation of simulations used and details should be placed in a Methods section at the end. I’m thinking of an organization like Sauermilch et al. (2021) in Nature Comms <https://doi.org/10.1038/s41467-021-26658-1>

The details regarding the model were moved from Data Availability statement to the “Numerical Simulations” subsection in the Results section and the Methods section.

Minor Comments

1. Line 189: I would add a topic sentence to this paragraph saying that differences in long-term SST gradients between the simulations and paleotemperature reconstructions may be attributed to the limited spatial distribution of available proxies (the topic of this paragraph is unclear from the first sentence so the transition from the last paragraph to this one is difficult to understand)

An introductory sentence to bridge the paragraphs was added. See Lines 203-205

2. Lines 234-239: Split this into two sentences at “which”

Done.

Reviewer #3 (Remarks to the Author):

The authors have done a thorough job of addressing my concerns. I especially welcome the expanded discussion on the model's paleogeography and recommend the publication of the manuscript.

We thank the reviewer for the gracious comments about the manuscript and are glad to hear that s/he has recommended the manuscript for publication in *Nature Communications*.

References

Adam, O., Farnsworth, A., & Lunt, D. J. (2022). Modality of the Tropical Rain Belt Across Models and Simulated Climates. *Journal of Climate*, 1–35. <https://doi.org/10.1175/JCLI-D-22-0521.1>